# Research on Vibration Energy Harvester Based on Two-Dimensional Acoustic Black Hole

**DOI:** 10.3390/mi14030538

**Published:** 2023-02-25

**Authors:** Chunlai Yang, Yikai Yuan, Hai Wang, Ye Tang, Jingsong Gui

**Affiliations:** 1School of Mechanical Engineering, Anhui Polytechnic University, Wuhu 241000, China; 2Anhui Key Laboratory of Advanced Numerical Control & Servo Technology, Wuhu 241000, China; 3Department of Mechanics, Tianjin University, Tianjin 300350, China; 4Wuhu Ruilong Robot Technology Co., Ltd., Wuhu 241000, China

**Keywords:** ABH, energy focus effect, vibration energy harvesting, piezoelectric

## Abstract

The wave energy focus effect of an acoustic black hole (ABH) is used for broadband vibration energy harvesting and boosts the harvested power. A vibration energy harvester based on two-dimensional ABH is proposed in this study, which consists of a rectangle plate with 2-D ABH and PZT film attached. The structure of ABH was designed and analyzed based on numerical simulation. The optimal parameters of the ABH were obtained, such as the power index, truncation thickness, cross-sectional length, and round table diameter, which were 3, 0.4 mm, 40 mm, and 24 mm, respectively. The quadratic velocity of the plate surface with ABH is up to 22.33 times that of a flat plate, and PZT film adheres to the corresponding positions of the ABH structure and plate structure, respectively. In the same condition, the average output power of a PZT with an ABH structure is higher than that of a flat plate under the same excitation-vibration condition.

## 1. Introduction

With the continuous development of Internet of Things (IoT) technology, low-power electronic products, such as wireless sensor networks (WSN) and portable wearable devices, have been widely used in recent years. Traditional battery methods are usually applied to low-power devices’ power supplies. The limited service life of batteries makes low-power electronic devices hard to maintain, especially when deployed in a remote area or harsh environment.

Various forms of energy sources widely exist in the natural environment, such as solar, wind, vibration, tidal, etc. Mechanical vibration energy is the most common energy source, which exists in production workshops, transportation such as vehicles and ships, construction such as tunnels and bridges, and human movement. A vibration energy harvester (VEH) [1,2,3] is an alternative battery power supply method, and it converts vibration energy into electric energy. VEH can be divided into three types according to their operation principle, such as piezoelectric, electrostatic, or electromagnetic energy harvesters. The piezoelectric VEH is most commonly used due to its simple structure and high electromechanical conversion efficiency. However, traditional piezoelectric VEHs with linear cantilever structures have narrow operating frequency bands and low harvesting efficiency.

Nonlinear methods can be applied in VEHs for frequency broadening and harvesting efficiency improvement [4,5,6], which consist of nonlinear structures and external nonlinear forces. A. J. Sneller [7] realized the transition from a monostable system to a bistable system by introducing an axial prestress beam in the vibration energy harvester. When the center mass increases from 0 to 6 μm, the output power of the energy harvester increases from 0.439 μW to 70.1 μW. M. Darakhshani [8] developed a bistable buckled structure by connecting the main beam and two cantilever beams with a torsion beam. The experimental results show that the main beam can realize bistable vibration under low-frequency excitation. A two-degree-of-freedom linear VEH with a stopper was proposed by M. A. Halim [9]. Upon excitation, the dynamic magnifier causes a mechanical impact on the base stopper and transfers a secondary shock to the energy harvesting element, resulting in an increased strain in it and triggering a nonlinear frequency up-conversion mechanism. Therefore, it generates almost four times larger average power and exhibits over 250% wider half-power bandwidth than those of its conventional 2-DOF counterpart without a stopper. The investigated system consisted of a bluff body in the form of a unique geometry. Ambrożkiewicz, B [10,11] analyzes the energy efficiency of a Micro Fiber Composite (MFC) piezoelectric system. The shape used, which is a hybrid combining square and circular cross-sections, is an example of a solution between galloping and the VIV phenomena. According to the obtained results, the high system efficiency in the whole range of flow velocities was obtained only for the measurement point for f = 4 Hz and the smallest mass of the bluff body.

The traditional nonlinear broadband piezoelectric type energy harvester has some limitations, such as a complex structure, miniaturization difficulty, and a high cost.

A VEH based on a two-dimensional acoustic black hole (ABH) is proposed in this study, which consists of a rectangle plate with a 2-D ABH and PZT film attached. The operation bandwidth broadened, and acquisition efficiency improved can be realized due to the focus effect of ABH. The influence of the ABH parameters is analyzed by numerical simulation, and optimal parameters are obtained. The performance characteristics of the vibration energy harvester, such as the broadband, the energy focalization effect, and the output voltages, are discussed.

## 2. Structural Design of 2-D ABH

### 2.1. ABH Theory

In a one-dimensional uniform dielectric beam, an acoustic black hole (ABH)’s wave resistance and energy concentration effects appear when the bending wave passes through a section that cuts the structure’s thickness according to a decreasing power-law relationship. Assume that the power-law expression of the variable thickness section is:(1)$$H\left(x\right)=\varepsilon X^{m}$$
where *H* is the thickness of the beam, *X* is the distance to the tip of the power-law curve, *ε* is a constant coefficient, *m* is the power exponent under a positive rational number, and *m* ≥ 2. The thickness of truncation (*h*_1_) cannot be zero during the actual manufacturing. The expression of the truncation thickness *h*_1_ at the center of the actual 2-D ABH structure is:(2)$$h\left(x\right)=\varepsilon x^{m}+h_{1}$$

Assuming that the thickness of the beam is constant, which is *H*, and the length of the ABH section is *L*, then *L* and *ε* are:(3)$$L=\sqrt[{m}]{\frac{H-h_{1}}{\varepsilon }}$$
(4)$$\varepsilon =\frac{H-h_{1}}{L^{m}}$$

Substituting the coefficient *ε* into Equation (2):(5)$$h\left(x\right)=\frac{H-h_{1}}{L^{m}}x^{m}+h_{1}$$

Figure 1 clearly shows the power function relationship of the section thickness; *x* represents the coordinate in the length direction of ABH.

The center of the actual two-dimensional ABH structure will form a circular platform with a truncated thickness of *h*_1_. The so-called two-dimensional ABH is the area formed by the one-dimensional ABH section with *x* = 0 as the circle’s center and one rotation. Therefore, the section still follows the power-law expression. For the thin plate with changing section thickness, its bending wave equation is expressed as:(6)$$\nabla^{2}\left(D\nabla^{2}w\right)-\left(1-\mu \right)\left(\frac{\partial^{2}D}{\partial x^{2}}\frac{\partial^{2}w}{\partial x^{2}}-2\frac{\partial^{2}D}{\partial x\partial y}\frac{\partial^{2}w}{\partial x\partial y}+\frac{\partial^{2}D}{\partial y^{2}}\frac{\partial^{2}w}{\partial y^{2}}\right)=\rho h\frac{\partial^{2}w}{\partial t^{2}}$$

When the moment of inertia and shear effect of the thin plate is not considered, the wave number *k* is obtained, and then the phase velocity *c* = *ω*/*k* of the bending wave is obtained. When the thickness of the section strictly follows the change law of the power function and the material is uniform, that is,
(7)$$c=\sqrt{\omega \left(\frac{H-h_{1}}{L^{m}}x^{m}+h_{1}\right)}{\left(\frac{E}{12\rho \left(1-\mu^{2}\right)}\right)}^{\frac{1}{4}}$$

Under ideal conditions, *h*_1_ = 0 and when *m* ≥ 2, as the thickness of the thin plate section decreases, the wave velocity of the flexural wave gradually decreases to 0, the wave amplitude increases slowly, and the flexural wave is concentrated at the center of the ABH, resulting in an energy focus effect.

### 2.2. Structural Design of 2-D ABH

A rectangular plate is designed, and the material of the plate is aluminum. As shown in Figure 2, a 2-D ABH is integrated at the position of *x* = 250 mm on the rectangle plate. The left boundary of the plate is fixed, and a 2 N point load with sinusoidal excitation is applied to the free right side of the plate.

The specific material properties and geometric dimensions are shown in Table 1. The thickness of the ABH changes according to the power function law. Since it is difficult to reduce the thickness to 0 due to the processing conditions, the truncated thickness is taken as 0.4 mm, so the central area of the ABH is a cylinder with a diameter of d and a height of 0.4 mm. Referring to Formula (5), the ABH structure parameters are shown in Table 2. The structure diagram is shown in Figure 3.

As shown in Figure 4, the PZT is pasted on the back of the central part of the ABH structure. PZT-5H can be selected as the PZT material, and the shape is round and leads out the circuit to connect the load on the positive and negative electrodes of the PZT. This is the vibration energy harvester based on the two-dimensional acoustic black hole designed in this paper.

### 2.3. Structure Analyses of 2-D ABH

#### 2.3.1. The Energy Concentration Effect of 2-D ABH

Based on the COMSOL modal analysis for flat plate and plate with ABH structure (ABH plate) to obtain the characteristic frequencies. Some frequencies are shown in Table 3.

It can be seen from Table 3 that the mode of the ABH plate has changed greatly and the characteristic frequency has decreased. It is calculated that the number of characteristic frequencies of the plate within 5000 Hz is 31, while the ABH plate is 36, which shows that the number of modes of the ABH structure has increased, as shown in Figure 5. It can be seen that the energy concentration effect of the ABH structure is not obvious under the mode of lower characteristic frequency, but with an increase in characteristic frequency, the ABH structure shows a very strong energy accumulation effect.

The frequency domain analysis is carried out in the range of 10–5000 Hz, and the quadratic velocity of the upper surface of the ABH region is compared with that of the corresponding region of the flat plate, as shown in Figure 6. It can be seen that the characteristic frequency of an ABH plate is reduced, and additional characteristic frequencies have been generated near 2800 Hz and 4000 Hz. At the same time, it can be seen that the advantages of the quadratic velocity of the ABH structure are not obvious compared with that of the flat plate before 1000 Hz, but after that, the quadratic velocity is greater. In the 5000 Hz frequency band, the average value of the quadratic velocity of the ABH plate is 7.24 mm/s, and the flat plate is 3.77 mm/s, increasing by 1.9 times.

#### 2.3.2. Influence of the ABH Power Index on Vibration

When the ABH length *L* and the truncation thickness *h*_1_ are fixed, the power index *m* affects the ABH section depth, as shown in Figure 7.

When *m* = 0, it is a flat plate. With the gradual increase in *m*, the local area of the section will have a smaller curvature radius. Here, *m* is taken as 2–20, and the step size is taken as 3. Table 4 shows the corresponding characteristic frequencies of high-order modes under different power exponents. It can be seen from Table 4 that the corresponding characteristic frequency of each mode shows a decreasing trend with the increase in power exponent *m* under the higher order mode, and this trend is gradual, that is, the characteristic frequency is closer to the increase in *m* value. Therefore, the vibration mode has a large deviation when the power index is low.

The transient simulation of the plate with ABH is carried out with a power exponent *m*, under a 2.4 kHz excitation frequency. The change in displacement at the center point of the circular plane of ABH over time is shown in Figure 8. The results show that, with the increase in power exponent *m*, the propagation velocity of the wave lags and decreases, so the corresponding wave amplitude increases. In addition, when *m* is larger, the increase in the focus effect of ABH does not change much.

#### 2.3.3. Influence of ABH Truncation Thickness on Vibration

The frequency response of the plate with ABH is numerically simulated with different truncation thicknesses *h*_1_, under an excitation frequency ranging from 10 Hz to 5000 Hz. To evaluate the energy focus effect of ABH, the logarithm of the ratio of the quadratic velocities of the ABH area and the flat area is recorded as *Ra*, which calls the focus ratio:(8)$$Ra=\log\frac{{\langle V^{2}\rangle }_{ABH}}{{\langle V^{2}\rangle }_{Flat}}$$

The focus ratio of the plate with ABH integrated under different truncation thicknesses *h*_1_ was carried out through numerical simulation, which is shown in Figure 9. The influence of truncation thickness is not different in the lower frequency region. However, the difference in the focus effect of ABH becomes significant when the excitation frequency is higher than 1500 Hz. When the plate with 2-D ABH is at the resonance frequency of 3150 Hz, the focus effect is increased with the smaller truncation thickness. The maximum *Ra* can reach one at the minimum truncation thickness, *h*_1_ = 0.4 mm.

#### 2.3.4. Influence of ABH Cross-Sectional Length on Vibration

The frequency response of the plate with ABH is numerically simulated with different cross-sectional lengths *L*, under excitation frequencies ranging from 10 Hz to 5000 Hz.

Figure 10 shows that the focus ratio *Ra* increases with the ABH cross-sectional length *L*, which means a better energy focus effect. However, for each cross-sectional length at a specific frequency, *Ra* will have a peak. For example, when the excitation frequency is 4000 Hz and *L* = 30 mm, the *Ra* can reach one. This may be due to the influence of the resonant frequency.

## 3. Optimization and Analysis

### 3.1. Optimization and Analysis of ABH Size

To achieve the best energy focus effect of ABH, four structural parameters are optimized in this paper, including the section length *L* of ABH, the power exponent *m*, the truncation thickness *h*_1_, and the diameter of the central circular platform *d*. Orthogonal test table L25 (56) is used for studying the effects of the influencing factors, as shown in Table 5.

According to the orthogonal test table, 25 experiments were completed. The kinetic energy density of the upper and lower round surfaces (diameter 15 mm) of the ABH structure was integrated. Twenty-five groups of experimental results, according to the integral results of Experiment 1, were normalized. The experimental results are shown in Table 6. It can be seen from the table that the order of the range values of each factor is: truncation thickness > cross-sectional length > round table diameter > power index. The order of the error square sum of each factor is the same. Therefore, the order of influence of ABH structure factors on the simulation results is the same.

In Table 6, the energy focus effect of Experiment 10 is the best, so the parameters of Experiment 10 are taken as the basis for the optimization of the structure parameters. Based on the parameters of Experiment 10, the parametric sweep is carried out on the section’s length, central cone diameter, and power exponent, respectively. The results were normalized according to Experiment 1, as shown in Figure 11.

To sum up, when the power index, truncation thickness, cross-sectional length, and round table diameter of the ABH are 3, 40 mm, 0.4 mm, and 24 mm, respectively, the ABH has the best energy focus effect. This group of structural parameters is taken as the parameters of Experiment 26.

### 3.2. Frequency Domain Analysis of Different Experiments

The simulation results of Experiment 10, Experiment 3, and the flat plate are compared to verify the above conclusions in the frequency domain analysis. Apply a simple harmonic excitation with an amplitude of 10 N at the right end of the cantilever plate, and the frequency range is 10–5000 Hz. Divide the frequency band into low frequency (10–1400 Hz) region, medium frequency (1400–3600 Hz) region, and high frequency (3600–5000 Hz) region. Compare the effects of the three groups of experiments with the quadratic velocities of the upper and lower surfaces of the ABH central platform (the circular area of the flat plate at the same position). The quadratic velocity at different frequencies is shown in Figure 12. The average value of the quadratic velocity at each frequency region is shown in Figure 13.

As can be seen from Figure 12, when the frequency is below 600 Hz, the experimental results of the three groups of parameters are close, and the energy focus effect of the ABH has almost disappeared. When the excitation frequency is higher than 600 Hz, the quadratic velocity of the plate with ABH is higher than that of the flat plate, which means the energy focus effect of the ABH has appeared. The best performance of the energy focus effect is Experiment 26. The quadratic velocity of the plate decreases rapidly when the excitation frequency is far away from the characteristic frequency of the plate, however, the reduction in speed of the ABH plate is smaller than that of the flat plate. It shows that the energy focus effect of ABH can be affected over a wider excitation frequency range.

The same conclusion can be drawn from Figure 13. In the low-frequency region, the average quadratic velocity of Experiment 26 is 1.79 times and 1.33 times higher than that of the flat plate and Experiment 3, respectively. In the medium-frequency region, the multiples are 7.6 and 3.05, respectively. In the medium-frequency region, the multiples are 7.6 and 3.05, respectively. In the high-frequency region, the multiples are 22.33 and 2.02, respectively. It can be seen that the ABH structure has obvious advantages over the flat plate, especially in the high-frequency region. In the frequency region of 10–5000 Hz, the average value of the quadratic velocity of Experiment 26 is increased by 5.03 times and 1.81 times compared with the flat plate and Experiment 3, respectively, showing the apparent advantages of the energy focus effect after the optimization of the ABH structure parameters. It shows that the energy focus effect is more significant with optimized ABH structure parameters.

### 3.3. Output Performance Simulation of VEH

Based on the above research, to further verify the advantages of the VEH composed of ABH. The frequency domain output performance of VEH is simulated. The circular PZT (PZT-5H) is pasted on the back of the central part of the ABH structure (corresponding to the position of the flat plate). An external resistance (1000 Ω) of the circuit on the positive and negative electrodes of the PZT. This is the VEH based on the two-dimensional acoustic black hole designed in this paper. The output performance of Experiment 26 VEH and the flat plate VEH is analyzed in the frequency domain by using simple harmonic excitation with an amplitude of 5 N. The simulation results are shown in Figure 14.

From Figure 14, in the low-frequency region, the average power of VEH in Experiment 26 is two times that of the flat plate. In the medium-frequency region, the multiple is 1.26. In the high-frequency region, the multiple is 4.25. It can be seen that the ABH structure has obvious advantages over the flat plate, especially in the high-frequency region. This is consistent with the result of the reaction in Figure 13, which further shows that the optimized ABH structural parameter VEH has a more excellent output performance.

## 4. Conclusions

The vibration energy harvester based on a two-dimensional ABH is studied in this article. Firstly, based on the working principle of ABH, the ABH effect and the influence of ABH structural parameters on the vibration are studied by simulation. Secondly, based on the orthogonal test method, the optimal ABH structure parameters are determined in the time domain analysis, namely, when the power index is 3, the truncated thickness is 0.4 mm, the cross-sectional length is 40 mm, and the round table diameter is 24 mm. Thirdly, the optimal ABH structure parameters are verified by frequency domain analysis. The results show that ABH structure can play a significant role in broadband, and the energy focalization effect is more significant after optimizing the ABH structure parameters. Finally, the output voltage contrast experiment shows that the output performance of VEH in Experiment 26 is significantly improved compared with that of the flat plate.

## Figures and Tables

**Figure 1 micromachines-14-00538-f001:**
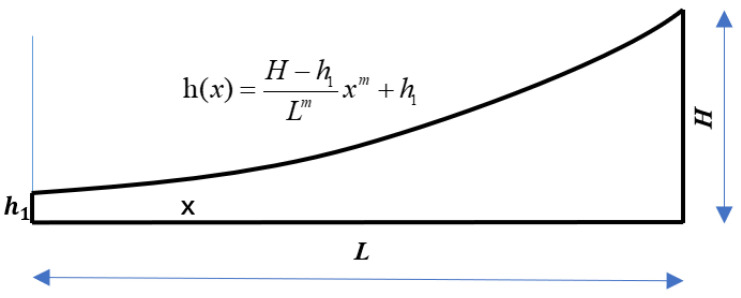
One-dimensional ABH shape outline.

**Figure 2 micromachines-14-00538-f002:**
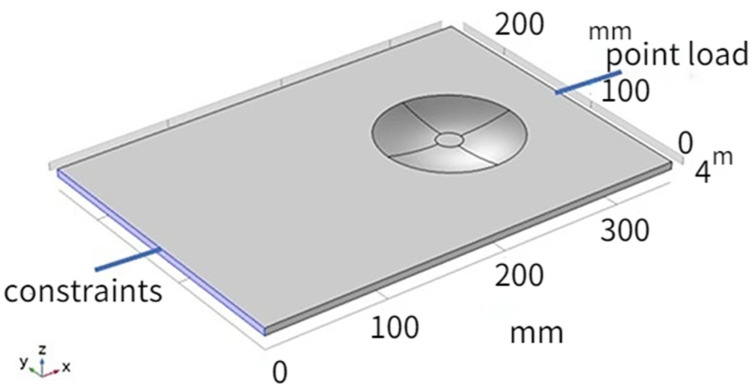
Rectangular plate with ABH structure.

**Figure 3 micromachines-14-00538-f003:**
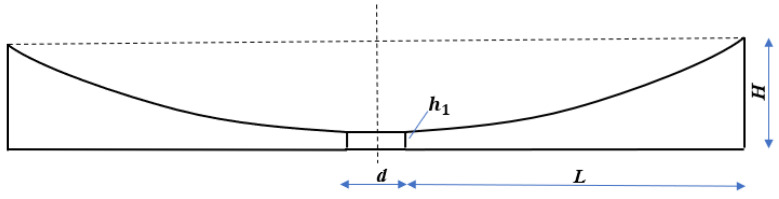
Two-dimensional ABH structure.

**Figure 4 micromachines-14-00538-f004:**
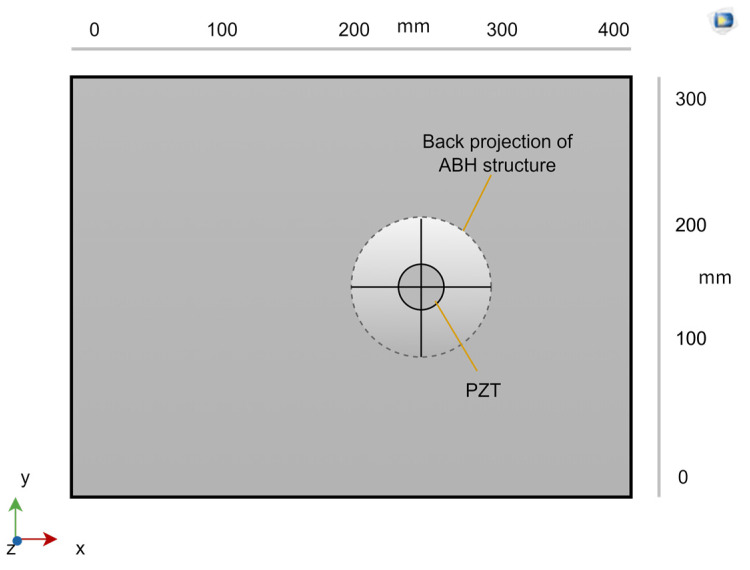
ABH board with PZT on the back.

**Figure 5 micromachines-14-00538-f005:**
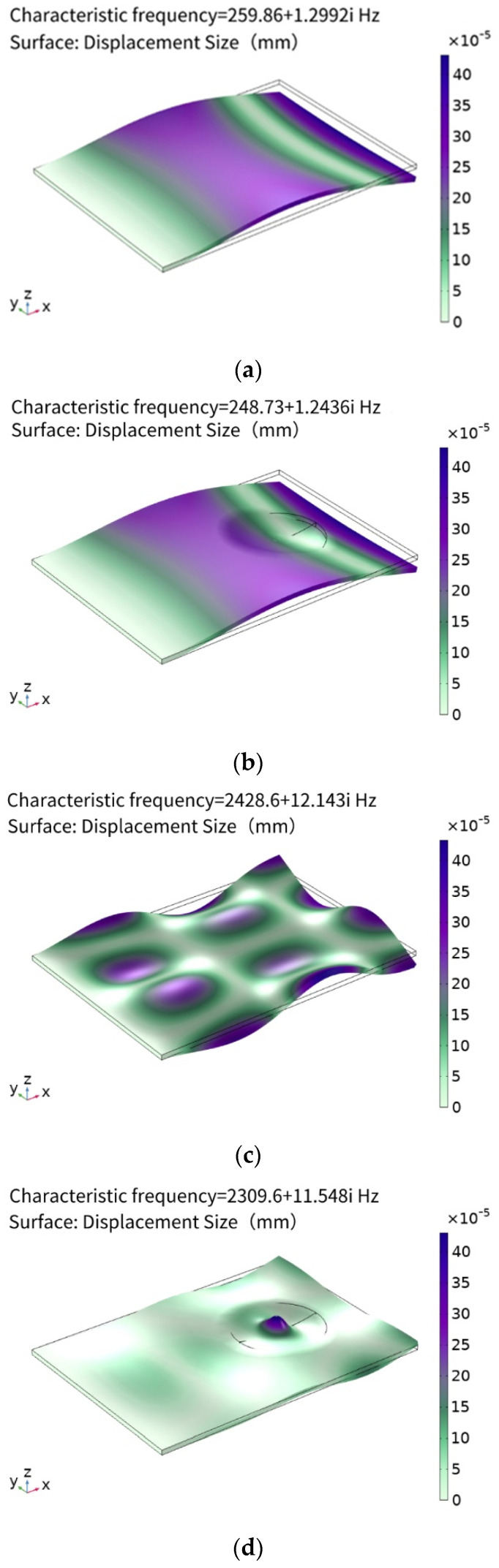
Modal comparison between flat plate and ABH plate. (**a**) Flat plate characteristic frequency 259.8 Hz, (**b**) ABH plate characteristic frequency 248.7 Hz, (**c**) flat plate characteristic frequency 2428.6 Hz, (**d**) ABH plate characteristic frequency 2309.6 Hz, (**e**) flat plate characteristic frequency 4721.5 Hz, and (**f**) ABH plate characteristic frequency 4055.3 Hz.

**Figure 6 micromachines-14-00538-f006:**
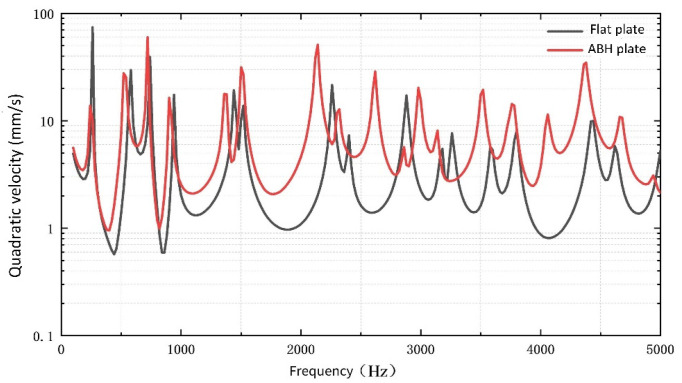
The quadratic velocity of flat plate and ABH plate at different frequencies.

**Figure 7 micromachines-14-00538-f007:**
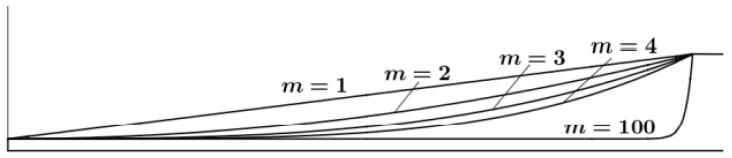
Power law section with a different power index.

**Figure 8 micromachines-14-00538-f008:**
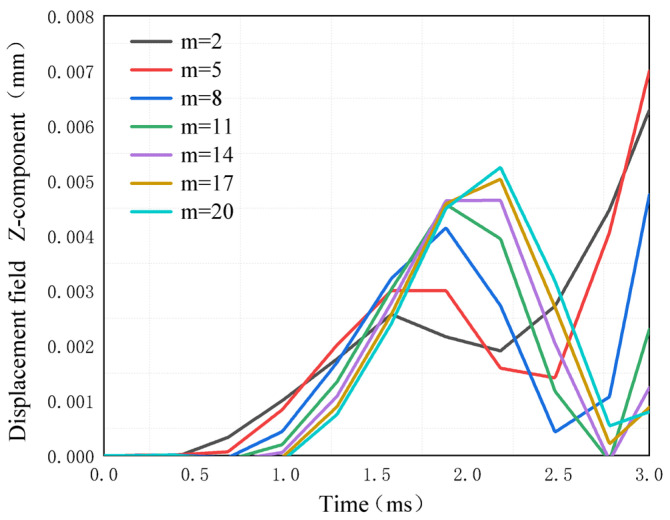
Displacement changes at ABH center point with a different power index.

**Figure 9 micromachines-14-00538-f009:**
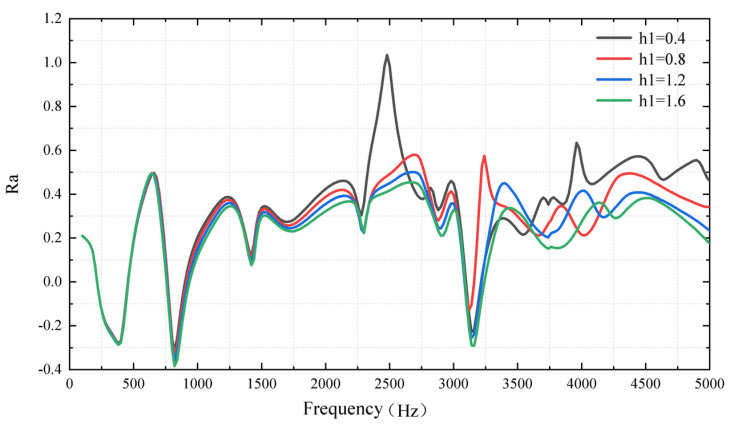
The ratio of quadratic velocity under different truncation thicknesses of ABH.

**Figure 10 micromachines-14-00538-f010:**
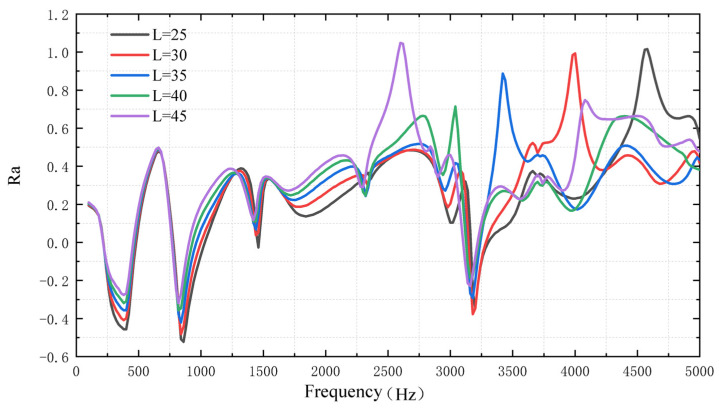
The ratio of quadratic velocity under different cross-sectional lengths of ABH.

**Figure 11 micromachines-14-00538-f011:**
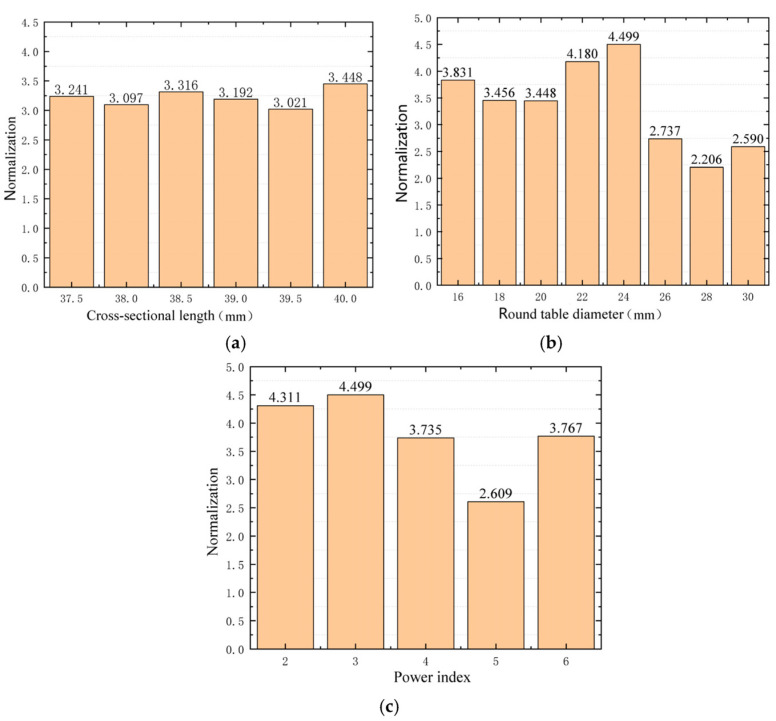
Parametric sweep results of ABH structural parameters. (**a**) Normalization corresponding to different cross-sectional length, (**b**) Normalization corresponding to different round table diameter, (**c**) Normalization corresponding to different power index.

**Figure 12 micromachines-14-00538-f012:**
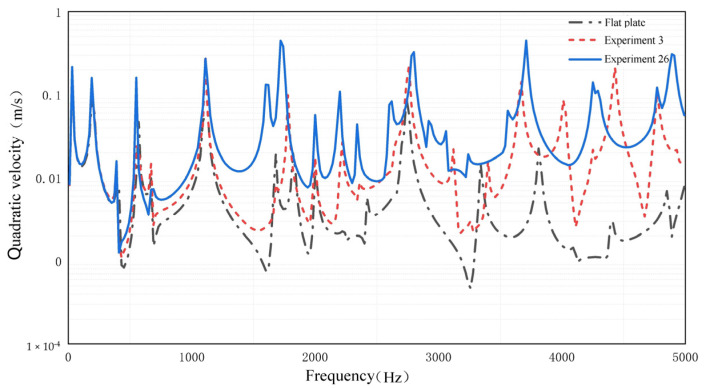
The Quadratic velocity of three experiments.

**Figure 13 micromachines-14-00538-f013:**
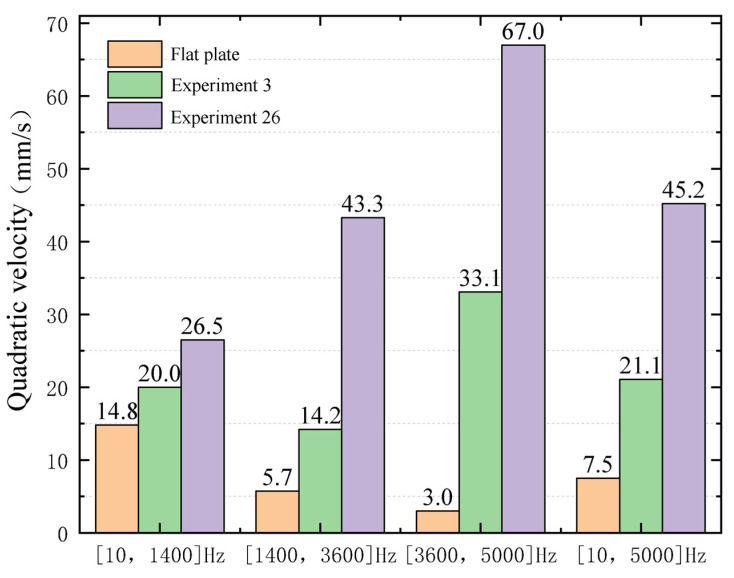
The average quadratic velocity of three experiments under different frequency regions.

**Figure 14 micromachines-14-00538-f014:**
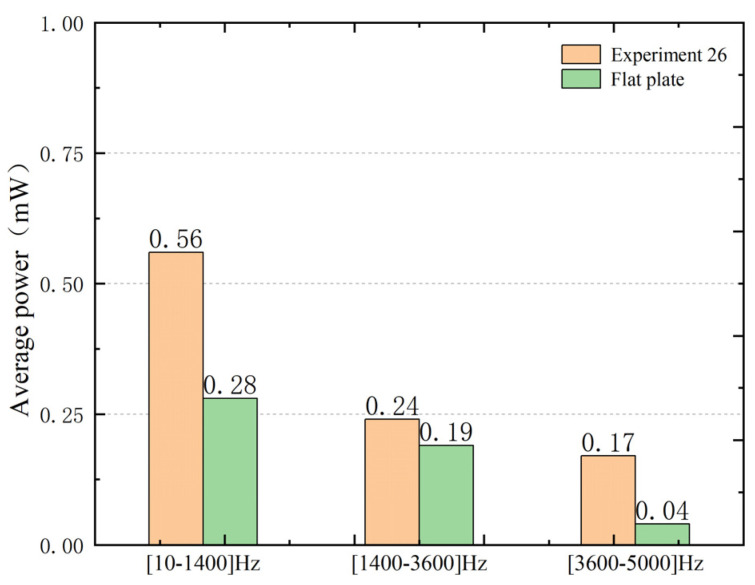
The output average power of the flat plate VEH and Experiment 26 VEH under different frequency regions.

**Table 1 micromachines-14-00538-t001:** Material properties and geometric parameters of the aluminum sheet.

Parameter	Numerical Value
Elastic Modulus (GPa)	70
Density (kg/m^3^)	2700
Poisson’s Ratio	0.3
Mechanical Damping	0.005
Length/mm	400
Thickness/mm	300
Width/mm	6

**Table 2 micromachines-14-00538-t002:** ABH geometric parameters.

Parameter	Numerical Value
Power Exponent m	2
Truncation Thickness *h*_1_/mm	0.4
Section Length *L*/mm	45
Round Platform Diameter *d*/mm	20

**Table 3 micromachines-14-00538-t003:** Comparison of characteristic frequencies of flat plates and ABH plates.

Order	Flat Plate	ABH Plate
First order	42.0 Hz	43.7 Hz
Second order	131.3 Hz	127.4 Hz
Third order	259.9 Hz	248.7 Hz
Sixteenth order	2397.8 Hz	2288.1 Hz
Seventeenth order	2428.6 Hz	2309.6 Hz
Eighteenth order	2604.4 Hz	2540.5 Hz
Twenty-nine order	4432.4 Hz	3967.1 Hz
Thirty order	4626.1 Hz	4049.8 Hz
Thirty-one order	4721.5 Hz	4055.3 Hz

**Table 4 micromachines-14-00538-t004:** Characteristic frequencies of cantilever plates with different power exponents.

*m*	Fifth-Order	Sixth-Order	Seventh-Order	Eighth-Order
2	720.21 Hz	905.74 Hz	908.66 Hz	1298.7 Hz
5	730.57 Hz	896.69 Hz	901.95 Hz	984.06 Hz
8	681.53 Hz	753.68 Hz	894.95 Hz	900.05 Hz
11	582.73 Hz	747.72 Hz	892.76 Hz	898.76 Hz
14	543.20 Hz	748.26 Hz	890.99 Hz	897.36 Hz
17	535.90 Hz	749.05 Hz	888.52 Hz	895.52 Hz
20	534.09 Hz	749.73 Hz	884.66 Hz	892.29 Hz

**Table 5 micromachines-14-00538-t005:** Orthogonal test table of ABH structure.

Serial Number	1	2	3	4
Factor	Power Exponent *m*	Section Length *L* (mm)	Truncated Thickness *h*_1_ (mm)	Diameter of Cone *d* (mm)
Level 1	2	2	0.4	15
Level 2	3	3	0.6	20
Level 3	4	4	0.8	25
Level 4	5	5	1.0	30
Level 5	6	6	1.2	35

**Table 6 micromachines-14-00538-t006:** Visual analysis of ABH structural parameters.

Factor	Power Index	Truncation Thickness	Cross-Sectional Length	Round Table Diameter	Simulation Result
Experiment 1	1	1	1	1	1.000
Experiment 2	1	2	2	2	1.684
Experiment 3	1	3	3	3	0.871
Experiment 4	1	4	4	4	0.740
Experiment 5	1	5	5	5	0.633
Experiment 6	2	1	2	3	1.145
Experiment 7	2	2	3	4	0.882
Experiment 8	2	3	4	5	0.979
Experiment 9	2	4	5	1	0.175
Experiment 10	2	5	1	2	3.448
Experiment 11	3	1	3	5	1.547
Experiment 12	3	2	4	1	0.547
Experiment 13	3	3	5	2	0.335
Experiment 14	3	4	1	3	3.000
Experiment 15	3	5	2	4	1.158
Experiment 16	4	1	4	2	0.493
Experiment 17	4	2	5	3	0.418
Experiment 18	4	3	1	4	3.445
Experiment 19	4	4	2	5	1.345
Experiment 20	4	5	3	1	0.416
Experiment 21	5	1	5	4	0.469
Experiment 22	5	2	1	5	0.563
Experiment 23	5	3	2	1	2.155
Experiment 24	5	4	3	2	0.930
Experiment 25	5	5	4	3	0.450
Mean value 1	0.986	0.931	2.291	0.859	
Mean value 2	1.326	0.819	1.497	1.378	
Mean value 3	1.317	1.557	0.929	1.177	
Mean value 4	1.223	1.238	0.642	1.339	
Mean value 5	0.913	1.221	0.406	1.013	
Range	0.413	0.738	1.885	0.519	
Error square sum	0.737	1.681	11.418	0.959	

## Data Availability

Not applicable.

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
