# Peer review of "Research on Vibration Energy Harvester Based on Two-Dimensional Acoustic Black Hole"

_micromachines, 2023, doi:10.3390/mi14030538_

Round 1

Reviewer 1 Report

Comments and Suggestions for Authors:

1.       The Abstract must be definitely shortened, please focus only on the novelties of your idea. I would keep the part starting from: “The wave energy focus effect…”.

2.       Over the manuscript I have found some punctuations and typos. Please revise the manuscript.  

3.       The good idea is to expand the bibliography with the part of applications of aero-flow nonlinear piezoelectric energy harvesting systems discussing galloping and VIV phenomenon. I suggest following papers for citation, but it is not compulsory:

·       Ceramic-based piezoelectric material for energy harvesting using hybrid excitation, Materials, 2021, 14(19), 5816

·       Performance analysis of piezoelectric energy harvesting system, Advances in Science and Technology Research Journal, 2022, 16(6), pp. 179-185

4.       In the end of the Introduction, please add the Remainder of the paper, what is discussed in the paper.

5.       Figures with plots are too small, please magnify them.

6.       Table 3, in what range of frequencies are you going to operate?

7.       What software do you use for simulation?

8.       What are your next steps of your research? Are you going to test the physical system?

After improving above described issues in the paper I’s like to sign my review report.

Author Response

Dear reviewer:

     Thank you for your decision and constructive comments on my manuscript. We have carefully considered the suggestion of Reviewer and make some changes. We have tried our best to improve and made some changes in the manuscript. Please find my itemized responses in attachment and my corrections in the re-submitted files.

Thanks again! 

Reviewer 2 Report

Review of “Research on Vibration Energy Harvester Based on Two-dimen-sional Acoustic Black Hole”

Chunlai Yang, Yikai Yuan, Hai Wang,Ye Tang, Jingsong Gui

This paper presents the optimization of an acoustic black hole structure for energy harvesting applications. The information presented in the paper is useful but needs some improvements.

Please add additional background information related to the use of ABH for energy harvesting structures.

Please check the source describing a 70.1 Watt power output (this seems to high) line 53, page 2.

I would move Figure 3 earlier in the text so that the readers get a clear idea of the concept. Also, please indicate where the PZT material would be place as well as the possible electrodes. In Figure 3, it is hard to see the numbers, please make those bigger.

The dimensions presented in this study are large for applications made with mciroelectronci techniques, such as MEMS. Please comment on this and what other possible fabrication methods could be used.

Add space between number and units such as 0.4 mm, 40 mm, instead of 0.4mm, 40mm.

Fix this sentence in page 1, line 40:

“It will be an alternative power supply method to the battery, by converting vibration energy into electrical energy  through an appropriate structure, which calls Vibration Energy Harvester (VEH)[1-3].”

Spell out ABH before its first use in the narrative in line 65, page 2. It is already mentioned in the abstract but should be spelled out again in the narrative.

I am not sure what it is meant by “truncation” in line 80, page 2. In addition, this sentence needs to be fixed as it is missing a verb:

“The thickness of truncation cannot precisely reduce to zero during the actual 80 manufacturing, which will leave a truncation at the end of the ABH.”

What is the loss factor shown in Table1 in this context?

Remove this sentence (it seems like it is just a place holder):

“This section may be divided by subheadings. It should provide a concise and precise description of the experimental results, their interpretation, as well as the experimental 134 conclusions that can be drawn.”

What are the vibration modes shown in Table 3? Where are the low order vibration frequencies? Comment on these.

There is mention of experiment 26 but table 5 only lists 25. What is experiment 26?

Please make Figures 9 and 10 bigger. They are hard to read.

What is the significance of using 5 Newtons to drive the vibration? Does that corresponded to a physical vibration force or any application?

Author Response

(The authors gave the same response as above.)

Reviewer 3 Report

Based on a careful analysis, I can formulate the following remarks:

1) The aim of this article, based on the authors’ scrupulous investigations, is to propose and to analyze a vibration energy-harvester based on two-dimensional Acoustic Black Hole (ABH), which consists of a rectangle plate with 2-D ABH and PZT film attached.

2) The topic represents in my opinion a relevant approach of the proposed theme in the field, based on meticulous theoretical and investigations, correlated with experimental results.

3)  In comparison with other published material, the authors' contribution adds to the subject area a new approach/methodology, with several significant contributions.

They performed a scrupulous a theoretical analysis, validated by meticulous experimental investigations.

The authors used the wave energy focus-effect of Acoustic Black Hole (ABH) for broadband vibration energy-harvesting and boost the harvested power.

They designed and numerically simulated a structure of ABH, obtaining the optimal parameters of the ABH, such as power index, truncation thickness, cross-sectional length, and round table diameter are 3, 0.4 mm, 40 mm and 24 mm, respectively.

The quadratic velocity of the plate surface with ABH is up to 22.33 times compared to flat plate. In the same condition, the output average power of PZT with ABH structure is higher than flat plate under the same excitation vibration condition.

 Their approach is better than the other reported ones from the literature.

4) Nowadays, with the continuous development of Internet of Things (IoT) technology, low-power electronic products such as wireless sensor networks (WSN) and portable wearable devices have been widely applied. It is well-known fact that the traditional battery methods, applied in low-power electronic devices are hard to maintain, especially when deployed in a remote area or harsh environment.

Various forms of energy sources widely exist in the natural environment, such as solar, wind, vibration, tidal, etc. Mechanical vibration energy is the most common energy source, which exists in production workshops, transportation such as vehicles and ships, constructions such as tunnels and bridges, and human movement.

It will be an alternative power supply method to the battery, by converting vibration energy into electrical energy through an appropriate structure, which calls Vibration Energy Harvester (VEH), from where the piezoelectric operation principle one is analyzed by the authors.

The piezoelectric VEH is the most commonly used due to its simple structure and high electromechanical conversion efficiency. However, traditional piezoelectric VEHs with linear cantilever structures have narrow operating frequency bands and low harvesting efficiency. Nonlinear methods can be applied in VEHs for frequency broadening and harvesting efficiency improvement, which consist of nonlinear structures and external nonlinear forces. Some authors realized the transition from a mono-stable system to a bi-stable system by introducing an axial pre-stress beam in the vibration energy harvester. The experimental results show that the main beam can realize bi-stable vibration under low-frequency excitation. A two-degree-of-freedom linear VEH with a stopper was also proposed. Upon excitation, the dynamic magnifier causes a mechanical impact on the base stopper and transfers a secondary shock to the energy-harvesting element resulting in an increased strain in it and triggering a nonlinear frequency up-conversion mechanism.

Therefore, it generates almost four times larger average power and exhibits over 250% wider half-power bandwidth than those of its conventional 2-DOF counterpart without a stopper.

The traditional nonlinear broadband piezoelectric type energy-harvester has some limitations such as complex structure, miniaturization difficulty, and high cost.

The authors propose a VEH-based on two-dimensional ABH, which consists of a rectangle plate with 2-D ABH and PZT film attached.

The operation bandwidth broadened and acquisition efficiency improved can be realized due to the focus effect of ABH. The influence of the ABH parameters, also numerically simulated offered finally the optimal parameters of the proposed system/device.

It is know, that in the one-dimensional uniform dielectric beam, an acoustic black hole (ABH)'s wave resistance and energy concentration effects appear when the bending wave passes through a section that cuts the structure's thickness according to a decreasing power-law relationship.

The center of the proposed two-dimensional ABH structure will form a circular platform with a truncated thickness of h1. The so-called two-dimensional ABH is the area formed by the one-dimensional ABH section with x=0 as the circle's center and one rotation. Therefore, the section still follows the power-law expression.

The authors designed a rectangular Aluminum plate, which constitute the so-called 2D ABH. The thickness of the ABH changes according to the power function law. Since it is difficult to reduce the thickness to 0 due to the processing conditions, the truncated thickness is taken as 0.4 mm, so the central area of the ABH is a cylinder with a given diameter of d and a height of 0.4 mm.

The frequency response of the plate with ABH is numerically simulated with different truncation thicknesses h1, under excitation frequency ranging from 10Hz to 5000Hz. To evaluate the energy focus effect of ABH, the logarithm of the ratio of the quadratic velocity of the ABH area and the flat area was the Ra focus ratio, also numerically simulated for different h1 truncation thicknesses.

 The influence of truncation thickness is not different at the lower frequency region. However, the difference of focus effect of ABH becomes significant when the excitation frequency is higher than 1500 Hz. When the plate with 2-D ABH is at the resonance frequency 3150 Hz, the focus effect is increased with the smaller truncation thickness. For the h1=0.4 mm Ra became maximum.

The frequency response of the plate with ABH, numerically simulated with different cross-sectional lengths L, under excitation frequencies ranging from 10 Hz to 5000 Hz shows that the focus ratio Ra is increased with the ABH cross-sectional length L, which means a better energy focus effect. However, for each cross-sectional length at a specific frequency, Ra will have a peak. For example, when the excitation frequency is 4000Hz and L=30mm, the Ra can reach 1. This may be due to the influence of the resonant frequency, how considered that authors.

To achieve the best energy focus effect of ABH, the authors stated to optimize four structural parameters, namely the section length L of ABH, the power exponent m, the truncation thickness h1, and the diameter d of the central circular platform. In this sense, they used an orthogonal test table, performing 25 experiments, from where the best response case is considered such basis of the proposed optimization.

Finally, when the power index, truncation thickness, cross-sectional length, and round table diameter of the ABH are 3, 40 mm, 0.4 mm, and 24 mm, respectively, ABH has the best energy focus effect.

      The obtained performances were better than the literature reported ones.

5) In my opinion, the presented conclusions are suitable related to their research results and prove that they reached the proposed goal.

6) The references in my opinion are very appropriate and their number underlines the scrupulosity of the authors.

7) In this paper, the graphical illustration is well conceived and realized and consequently they contribute to a better understanding of the performed theoretical investigations as well as to underlining the experimental validation of the proposed methodology.

I express my sincerely hope that in the next period the authors intend continuing these useful theoretical and experimental investigations.

 I encourage publishing in a new contribution their further results.

Author Response

Dear reviewer:

     Thank you for your decision and constructive comments on my manuscript.  We have tried our best to improve and made some changes in the manuscript. Please find my corrections in the re-submitted files.Thank you for your affirmation of our work.Looking forward to more guidance in the future.

Thanks again! 

Round 2

Reviewer 1 Report

Accept in the present form

Reviewer 2 Report

I would like to thank the authors for their timely revisions. They have addressed all the reviewer's major concerns.

Minor revisions/text editing to consider before publication.

The should consistently use a space between numbers and units throughout the text.

Consistently capitalize the word Experiment in Experiment 26.